# Conditioner application improves bedding quality and bacterial composition with potential beneficial impacts for dairy cow's health

Lysiane Duniere,[1] Bastien Frayssinet,[1] Caroline Achard,[1] Eric Chevaux,[1] Julia Plateau[1]

**ABSTRACT** Recycled manure solids (RMS) is used as bedding material in cow housing but can be at risk for pathogens development. Cows spend several hours per day lying down, contributing to the transfer of potential mastitis pathogens from the bedding to the udder. The effect of a bacterial conditioner (Manure Pro, MP) application was studied on RMS-bedding and milk qualities and on animal health. MP product was applied on bedding once a week for 3 months. Bedding and teat skin samples were collected from Control and MP groups at D01, D51, and D90 and analyzed through 16S rRNA amplicon sequencing. MP application modified bacterial profiles and diversity. Control bedding was significantly associated with potential mastitis pathogens, while no taxa of potential health risk were significantly detected in MP beddings. Functional prediction identified enrichment of metabolic pathways of agronomic interest in MP beddings. Significant associations with potential mastitis pathogens were mainly observed in Control teat skin samples. Finally, significantly better hygiene and lower Somatic Cell Counts in milk were observed for cows from MP group, while no group impact was observed on milk quality and microbiota. No dissemination of MP strains was observed from bedding to teats or milk.

**IMPORTANCE** The use of Manure Pro (MP) conditioner improved recycled manure solids-bedding quality and this higher sanitary condition had further impacts on dairy cows' health with less potential mastitis pathogens significantly associated with bedding and teat skin samples of animals from MP group. The animals also presented an improved inflammation status, while milk quality was not modified. The use of MP conditioner on bedding may be of interest in controlling the risk of mastitis onset for dairy cows and further associated costs.

**KEYWORDS** RMS-bedding, mastitis, bacterial communities, teat skin, SCC

Bedding materials for dairy cows can be inorganic (sand) or organic such as sawdust or wood products. Recycled manure solids (RMS) is a type of organic bedding made from fresh manure and brought to at least 34% dry matter (DM) through removal of water using a screw, a roller press, or high-performance slurry separation equipment. Despite this process, several pathogens have been isolated from RMS bedding (1). For example, higher detection of *Salmonella* spp. and *Listeria monocytogenes* was observed in unused RMS compared to unused straw beddings (2). Due to high DM content, sand and sawdust are often considered to be "safer" bedding material; however, potential pathogenic bacteria such as coliforms, *Klebsiella,* and *Streptococcus* were found in both bedding materials in free-stall dairy farm (3). Clean stall bedding is very closely related to animal hygiene and bacterial exposure. Dairy cows lying time has been measured as 10– 13 h per day (4), and during this period, cows' teats are in direct contact with

Address correspondence to Lysiane Duniere, lduniere@lallemand.com, or Lysiane Duniere, lduniere@lallemand.com.

All authors are employees of Lallemand SAS.

See the funding table on p. 16.

bedding materials. Dirty bedding, associated with poor animal hygiene and soiled udder, has been recognized as a source of intramammary infections and an increasing risk of mastitis onset (5–8). The main pathogens responsible for mastitis are *Streptococcus*, *Staphylococcus*, *Escherichia*, *Klebsiella,* or *Corynebacterium* spp., but up to 24 bacterial genera have been identified from udder pathogenic isolates (9, 10). Mastitis is classified as clinical or subclinical depending on the visibility of mammary gland inflammation. Subclinical mastitis does not produce visible effects on milk quality but has important impacts on milk composition, mainly through an increase in somatic cell counts (SCC), while clinical mastitis is associated with abnormal milk detection of flakes and clots and visible pathological changes of the mammary tissue (red and swollen udder, fever, etc.) (11). Thus, bovine mastitis has, thus, a massive negative effect on animal well-being and also on farm economics due to treatment costs, reduction in milk production, and early culling (12–17). It is one of the most frequent diseases in the dairy industry and in a meta-analysis gathering 372 studies spanning the period 1967–2019, worldwide; the prevalence of sub-clinical mastitis was observed to be 42%, while clinical mastitis reached 15% (18).

Results are unclear about the specific impact of RMS bedding on mastitis occurrence in dairy cows. RMS was suggested to be associated with lower udder health (19) and increased sub-clinical mastitis risks (20) compared to other organic or inorganic beddings, while in other studies, no correlation was found between RMS bedding and sub-clinical mastitis (21), animal hygiene (22), and SCC in milk (23). Only a few studies have been published on the use of bedding conditioners to prevent mastitis. Chemical disinfectants can be used to limit bacterial counts in organic bedding and decrease their transfer onto teat skin (24, 25), but the effects are variable depending on bedding material, pathogens considered, and duration of antibacterial activity (26, 27). Moreover, some bedding conditioners such as hydrated lime [calcium hydroxide, $Ca(OH)_2$] are caustic and can cause animal and farm worker's skin damage.

The use of specific bacteria to drive the microbial populations through fermentation is widely used in agronomy [silage inoculant (28), composting process (29), etc.]. Bioremediation and ecological niche occupation with specific bacterial strains have also been successfully employed in soil depollution (30) or positive biofilm production in the food industry, for example (31, 32). The bedding conditioner tested in this study is composed of a cocktail of *Bacillus* and *Pediococcus* strains and enzymes aimed at promoting beneficial bacteria and limiting the presence of undesirable microorganisms through the development of a biofilm. *Bacillus* spp. produce a great variety of extra-cellular enzymes such as α-amylase, cellulase, and proteases involved in organic matter degradation (33). However, they are recognized as milk spoilage bacteria causing shelf-stability issues in dairy products including ropiness, sweet curdling, and off-flavors (34), and their presence or potential transfer in milk should be avoided carefully.

Providing a clean and safe standing and lying environment by controlling bedding microbial quality is, thus, a key step in mastitis control. Up to now, no study on the effects of bacterial bedding conditioner on dairy cows' bedding quality and further animal health has been published. The aim of this study was (i) to study the effect of a bacterial bedding conditioner (Manure Pro, Lallemand SAS, France), further identified as Manure Pro (MP) in the manuscript, on RMS-bedding physico-chemical parameters and bacterial populations; (ii) to evaluate the impacts of this bedding treatment on animal hygiene, health, and teat skin bacterial populations; and, finally, (iii) to assess the impact on milk quality.

## RESULTS

### Physico-chemical parameters and bacterial population of bedding samples

The two groups tended to differ between D01 and D51 (Period 1) with a decrease in DM observed in the Control (−10.39% DM), while the DM of the treated bedding slightly increased (+1.97% DM, Table 1). A tendency for a stronger pH increase was also noted for MP bedding during the same period (+0.20 and +0.55 pH unit for Control and MP,

**TABLE 1** Physico-chemical parameters of bedding samples measured at D01, D51, and D90 in Control and MP group (n = 6). Mean, SEM, and P-values associated during Period 1 (D0–D51) and Period 2 (D51–D90)

| | Control | | | MP | | | | P-values | |
|---|---|---|---|---|---|---|---|---|---|
| | D01 | D51 | D90 | D01 | D51 | D90 | SEM | Period 1 | Period 2 |
| DM | 61.01 | 50.62 | 54.83 | 57.61 | 59.58 | 50.71 | 1.199 | 0.053 | 0.130 |
| pH | 8.76 | 8.96 | 9.07 | 8.52 | 9.07 | 9.09 | 0.038 | 0.098 | 0.247 |
| OM | 39.33 | / | 34.84 | 33.76 | / | 37.84 | 1.258 | | 0.250 |

respectively). No significant change in the measured parameters was observed in Period 2.

Both experimental bedding groups were characterized by a stable relative abundance of the phyla Firmicutes (average of 33.03% ± 7.72%) and Proteobacteria (average of 26.75% ± 5.16%) over time and a transient decrease of Actinobacteria at D51 (Fig. S1A). Bacterial families associated with potential and recognized mastitis pathogens (i.e., Staphylococcaceae and Streptococcaceae) were identified in both Control and Manure Pro groups (Fig. S1B).

Bacterial richness did not vary significantly according to Time or Group [observed amplicon sequence variants (ASVs), $P > 0.05$ for both variables] although numerically higher richness was observed in MP samples (Fig. 1A). A transient decrease of Shannon diversity values was observed at 51 days in both Control and MP groups (Shannon indice, time effect, $P = 0.006$) (Fig. 1B). An almost significant Group effect was also observed ($P = 0.063$) with numerically greater Shannon diversity values for MP compared to Control samples highlighting the higher heterogeneity of bacterial populations in MP samples. There was a clear shift in of the bacterial community structure in all bedding samples over time (Fig. 1C, $P = 0.001$), while a Group effect could be observed only after 90 days ($P = 0.003$) between Control and MP samples.

Differential analysis was performed with Indicspecies R package which determines lists of ASVs that are more specifically associated to each experimental group through an Indicator Value (IndVal) index. The IndVal index is a combination of A and B components. Component "A" represents the *specificity* or *positive predictive value* of the species as an indicator of the target experimental group. A species with A = 1 is a good indicator of the experimental group as it is found only in this specific group. Component "B" is called the *fidelity* or *sensitivity* of the species as an indicator of the target experimental group. A B = 1 species indicates that this species is observed in all samples from the experimental group considered. In order to circumvent the issue of taxonomic multi-affiliation occurring for several species belonging to the *Streptococcus* or *Staphylococcus* genera, a decision was made to agglomerate ASVs at the genus level. Linear discriminant analysis (LDA) Effect Size (LEfSe) analysis, commonly employed in microbial ecology studies, was used as a complementary differential analysis to confirm the taxa identified through IndicSpecies analysis.

Similar numbers of taxa were identified as characteristics of each experimental group with the highest and lowest number of taxa identified in MP at D01 and D51 (36 and 12 taxa identified, respectively) (Table 2). Interestingly, taxa in the MP groups were identified as environmental nonpathogenic bacteria such as *Mesorhizobium* or commensal gut bacteria such as Lachnospiraceae NK3A20 group and Christensenellaceae R7 group, and no taxa containing potential pathogen were identified (Supplementary Material xlxs document).

On the contrary, taxa belonging to genera containing species responsible for mastitis were associated with the Control groups at different sampling times. *Streptococcus* and *Acinetobacter* were identified at D51, and *Staphylococcus* was also observed at D90 in all samples of the Control group of interest (B = 1). Other potential pathogens were associated with Control group at D01 or D51: *Fusobacterium* was only observed in D51 samples (A = 1), while *Legionella* was significantly associated with D01. LEfSe analysis (Fig. 2) identified *Enterococcus* in Control D01 samples and *Psychrobacter* and *Corynebacterium* in Control D90 while at D51, the only taxa associated with MP treatment was

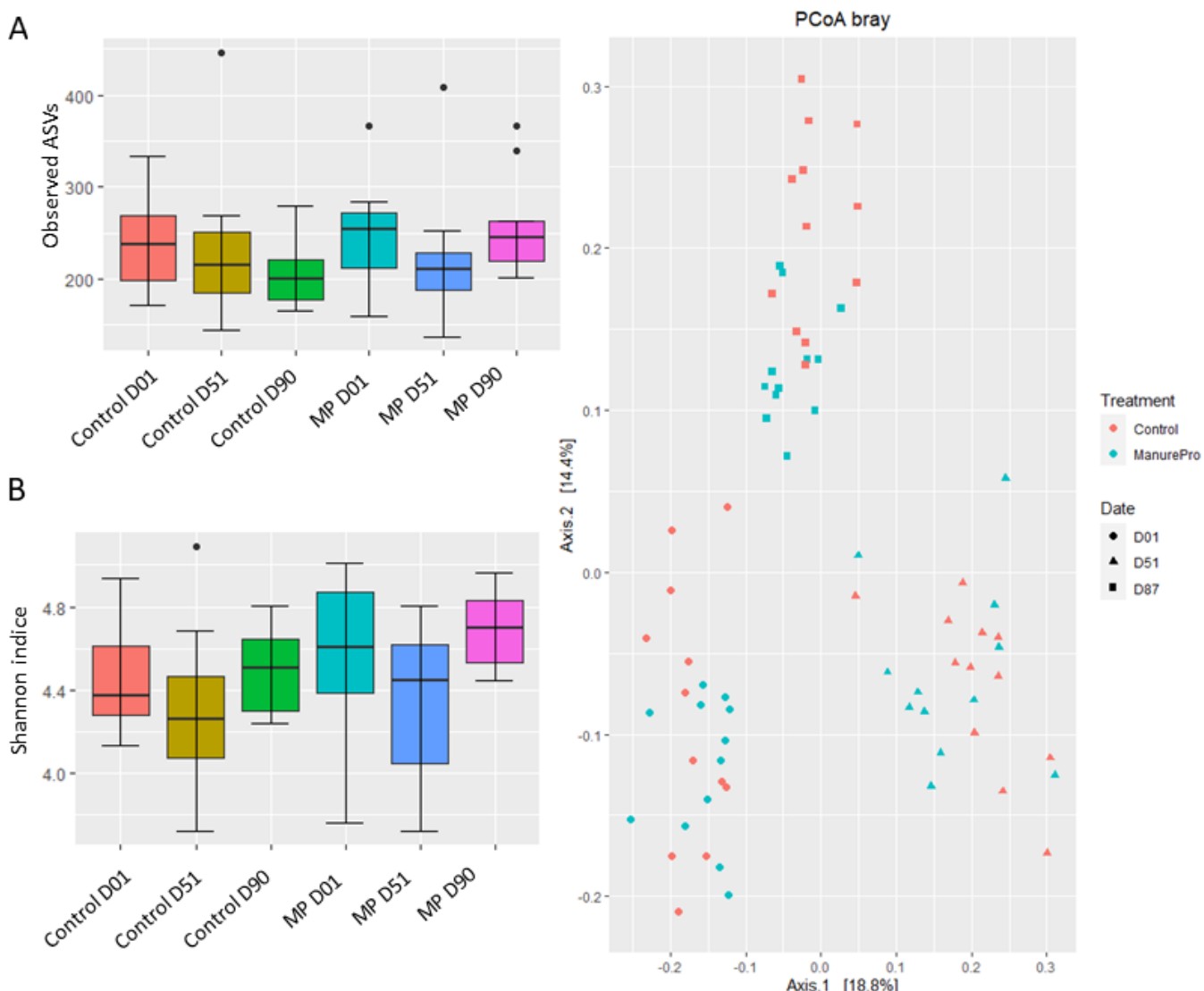

**FIG 1** Alpha diversity indices expressed through (A) observed ASVs and (B) Shannon indice, and beta diversity expressed through (C) PCoA plot based on Bray–Curtis distance for normalized abundance data of Control and Manure Pro bedding samples at D01, D51, and D90 ($n = 12$).

the commensal genus of *Glutamicibacter*. Notably, a taxa assigned to *Bacillus* genus was associated with Control D51 through Indicspecies analysis (A = 0.35, B = 0.92), while none was associated with MP groups.

The PICRUSt2 pipeline was used to predict the functional potential of the bedding microbiota using the MetaCyc pathway database. The comparisons between Control and MP samples considering each sampling time separately identified few significantly different potential metabolic pathways (Fig. 3, $P < 0.05$). No significant difference was observed between Control and MP at D01, while two pathways linked to biogenic amines production were predicted to be more abundant in Control than in MP samples at D51 and 7 pathways linked to several biological functions were identified in higher proportions in MP group at D90, notably a pathway of nitrifier denitrification which might be of interest in bedding management.

## Animal health and teat skin bacterial population

Animal hygiene was assessed through cleanliness score, concentration of teat skin culturable microflora (Table 3), and SCC in milks over the trial (Fig. 4). A significant Time

**TABLE 2** Number of taxa and taxa of potential pathogens identified through Indicspecies analysis, *P*-values associated and relative abundances observed for Control and MP beddings analyzed at D01, D51, and D90

| Group | Number of identified taxa | Taxa of potential pathogens | *P*-value | Control (%) | MP (%) |
|---|---|---|---|---|---|
| Control D01 | 28 | *Legionella* (A = 0.464; B = 0.5) | 0.024 | 0.06 | 0.04 |
| | | *Streptococcus* (A = 0.351; B = 1) | 0.001 | 5.66 | 3.49 |
| | | *Fusobacterium* (A = 1; B = 0.25) | 0.031 | 0.02 | 0 |
| Control D51 | 28 | *Acinetobacter* (A = 0.25; B = 1) | 0.008 | 12.5 | 11.43 |
| Control D90 | 13 | *Staphylococcus* (A = 0.593, B = 1) | 0.001 | 2.02 | 0.44 |
| MP D01 | 36 | NA | NA | | |
| MP D51 | 12 | NA | NA | | |
| MP D90 | 21 | NA | NA | | |

by Group interaction was observed for these three parameters. At D51, a significantly deteriorated hygiene was observed concomitantly to a numerically higher microbial concentration on teat skin of Control cows compared to MP animals.

SCC in milks was strongly decreased in MP group with significantly lower somatic cell concentration between D44 and D76 compared to milk from Control cows (Fig. 4). A highly significant effect of both Time and Group factors on SCC was also observed over the trial ($P < 0.0001$). Considering a 250,000 cells/mL threshold, 305 milk samples from cows in Control group have been measured above this limit during the trial for only 151 milk samples in MP group (Fisher test, $P < 0.0001$).

Finally, the number of mastitis suspicions based on milk conductivity above 90 mS/cm was not significantly different between Control and MP groups ($P > 0.05$, Table S1) either when cases were considered per week or over the entire experimental period. Although not significant, the total number of mastitis suspicion cases leading to an effective pathogen identification through on-farm culture plating was higher in Control than in MP group (57 cases in Control and 51 cases in MP group, respectively, $P = 0.66$), with a higher number of suspicion cases identified as environmental mastitis pathogens (35 cases in Control and 22 cases in MP group, respectively, $P = 0.143$).

As previously observed for bedding bacterial composition, the relative abundances of bacterial phyla in teat skin samples were quite stable over time in both Control and MP groups (Fig. S2A). Teat skin bacterial community was dominated by Firmicutes (38.76%

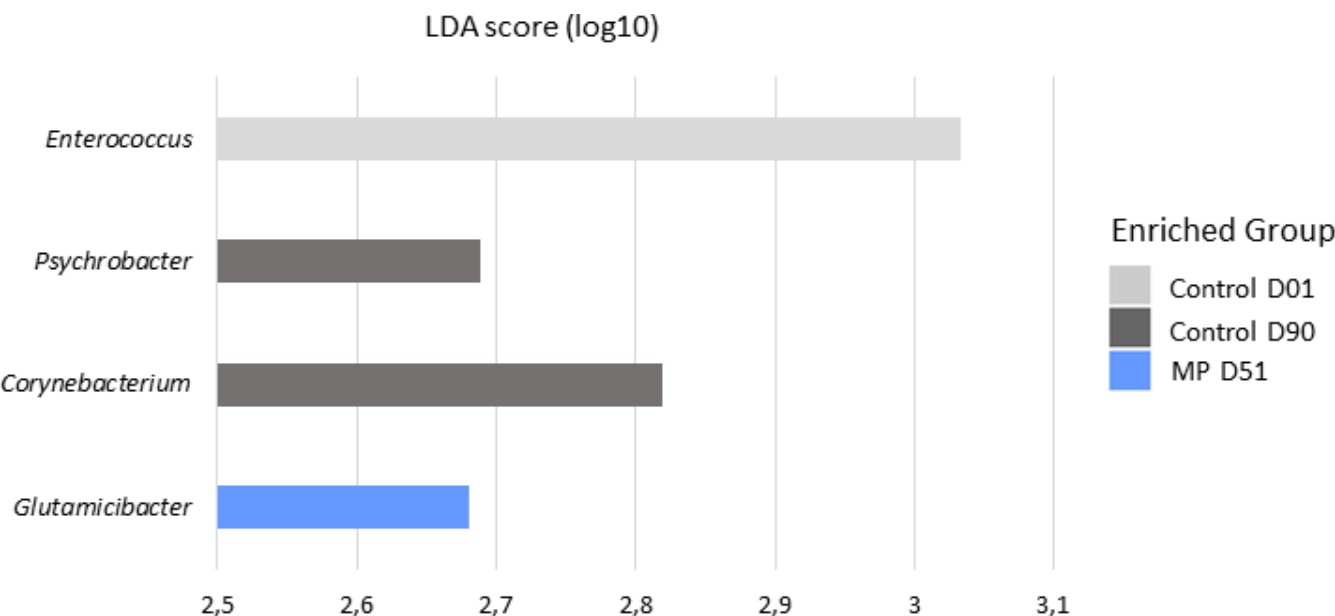

**FIG 2** LEfSe analysis identifying specific genera level on Control (gray bars) and ManurePro (blue bars) bedding samples at D01, D51, and D90 (LDA cuttoff 2.5, *P* < 0.05).

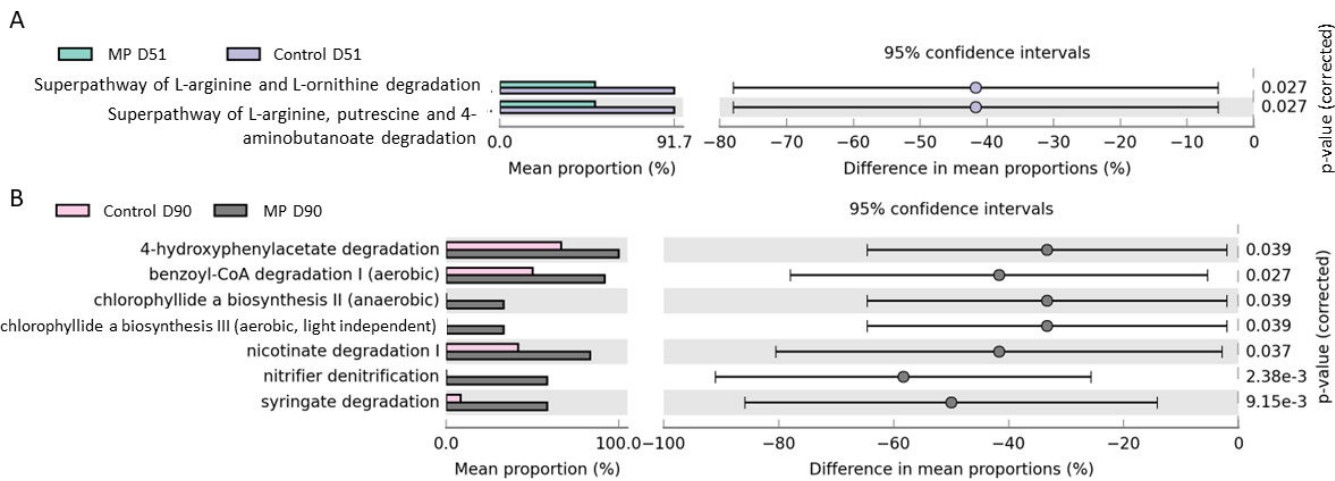

**FIG 3** Statistically different metabolic pathways identified in Control and Manure Pro bedding samples at (A) D51 and (B) D90 identified through PICRUSt2 pipeline prediction.

± 12.30%), followed by Proteobacteria (22.79% ± 7.79%), Bacteroidota (20.38% ± 7.39%), and Actinobacteria (17.45% ± 7.13%). At Family level, similar profiles were observed between Control and MP samples and an evolution over time was observed at D90 with an increase in Pseudomonadaceae (from 2.335% to 6.44%) and a decrease in Enterococcaceae (from 6.06% to 2.13%) and Streptococcaceae (from 4.10% to 1.16%) families (Fig. S2B). Bacterial families including potential and recognized mastitis pathogens such as Streptococcaceae (3.21% ± 4.57%) and Staphylococcaceae (2.55% ± 2.18%) were observed in Control and MP groups with variations over time. A significant interaction of Time × Group (observed ASVs, $P = 0.045$) and a significant effect of Time (Shannon, $P < 0.0012$) and of Group (Shannon, $P = 0.025$) were observed on alpha diversity indices, highlighting a transient decrease of bacterial diversity at D51 and an overall higher diversity of bacterial teat population in MP group at D01 (Fig. 5A and B). The temporal evolution of bacterial profiles of both Control and MP samples was also observed through beta diversity analysis (Fig. 5C, $P < 0.001$ for each sampling time). Notably, a significant difference was observed between MP and Control at each sampling time ($P_{D01} = 0.008$, $P_{D51} = 0.033$, and $P_{D90} = 0.003$).

Indicspecies analysis identified a higher number of taxa significantly associated with MP samples compared to Control samples at both D01 and D90 (Table 4). Interestingly, lactic acid bacteria were significantly associated to Control samples at D01 and *Bifidobacterium* at D90. Potential mastitis pathogens such as *Trueperella* was observed in almost all Control samples at D90. Although the mean relative abundance was higher in Control group (11.33%), MP samples were specifically associated with *Acinetobacter* at D51 as all those samples harbored this genus (B = 1). *Pseudomonas* was also particularly observed in MP samples at D90. No taxa associated to potential pathogens was observed in MP sample at D01 or in Control samples at D51. Interestingly, the same taxa of *Bacillus* associated to Control D51 in bedding samples was also observed in Control teat samples at D51 (A = 0.54, B = 0.95).

**TABLE 3** Animal hygiene parameters (cleanliness score and concentration of teat skin flora) observed in Control and MP cows at D01, D51, and D90 ($n = 20$)[a]

| | Control | | | MP | | | | P-values | | |
|---|---|---|---|---|---|---|---|---|---|---|
| | D01 | D51 | D90 | D01 | D51 | D90 | SEM | Time | Group | Time × Group |
| Cleanliness score/6 | 1.99[a] | 2.48[bA] | 2.74[b] | 2.11[a] | 2.10[aB] | 2.69[b] | 0.083 | <0.001 | 0.404 | 0.049 |
| Teat skin bacteria (log$_{10}$ CFU/mL) | 7.94[a] | 9.35[b] | 7.34[a] | 8.19 | 8.59 | 7.84 | 0.115 | <0.001 | 0.993 | 0.031 |

[a]Mean, SEM and p-values associated. Lower superscript letters indicates a significant difference across time and capital superscript letters a significant difference between groups.

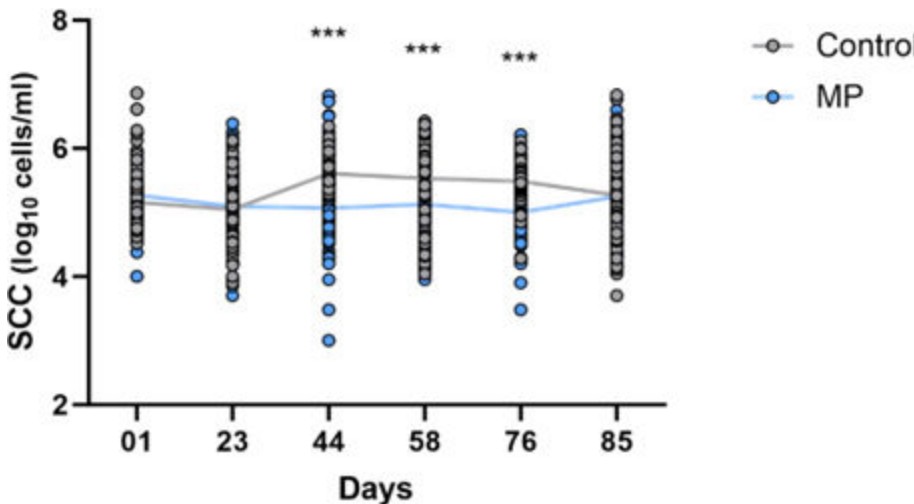

**FIG 4** $Log_{10}$ SCC/mL in milk collected from cows assigned to Control (gray, $n$ = 113) or Manure Pro (blue, $n$ = 115) group over the trial. *** indicates a post-hoc Sidak's $P$-value < 0.0001.

LEfSe analysis significantly associated *Streptococcus* and *Enterococcus* genera to Control D51 teat samples, while no taxa was associated with MP teat samples (Fig. 6).

## Milk production and quality

Milk production was followed for all enrolled animals in Control and MP groups by adding the quantity produced at each milking per day for each animal and averaged by month of treatment application (Month 1: D01–D30, Month 2: D31–D60; Month 3: D61–D90) (Fig. S3). No significant difference was observed for time, treatment, or the interaction of the two factors.

Milk quality was assessed through total culturable microbiota and aerobic spore concentration (Table 5). No significant effect of Group was observed on these parameters over time. Total milk bacteria increased up to 4.3 $log_{10}$CFU/mL at D90 in the Control group while it reached its highest concentration (4.51 $log_{10}$ CFU/mL) at D51 in MP. Spore counts were ranging between 10 and 14.06 spores/mL on average in Control group and between 12.65 and 15.59 in MP group.

## DISCUSSION

The use of RMS as bedding material has been increasing over the last years as an interesting valorization strategy for cattle organic wastes (i.e., fresh manure). Several studies were focused on the presence of potential pathogens in this environment but were mainly performed using cultural detection methods (1, 23), and very little data are available on the evolution of the overall bacterial populations through amplicon sequencing (35). Moreover, the interest of bedding conditioners to improve beddings quality has been poorly studied. To our knowledge, this study is the first one assessing the effect of a cocktail of bacteria (*Bacillus velezensis*, *Pediococcus acidilactici,* and *Pediococcus pentosaceus*) and enzymes on the quality of RMS-bedding with a focus on potential mastitis pathogens.

### MP application improved bedding physico-chemical and bacterial quality

The sanitary safety of RMS is based on the fact that microorganisms initially present in fresh manure will not develop or survive in the new environmental conditions prevailing in RMS (i.e., high DM and alkaline pH), thus limiting further risks for animal and human health (36). Physico-chemical values obtained for bedding in both groups were in line with literature (23, 37) with MP application leading to higher DM and pH values than

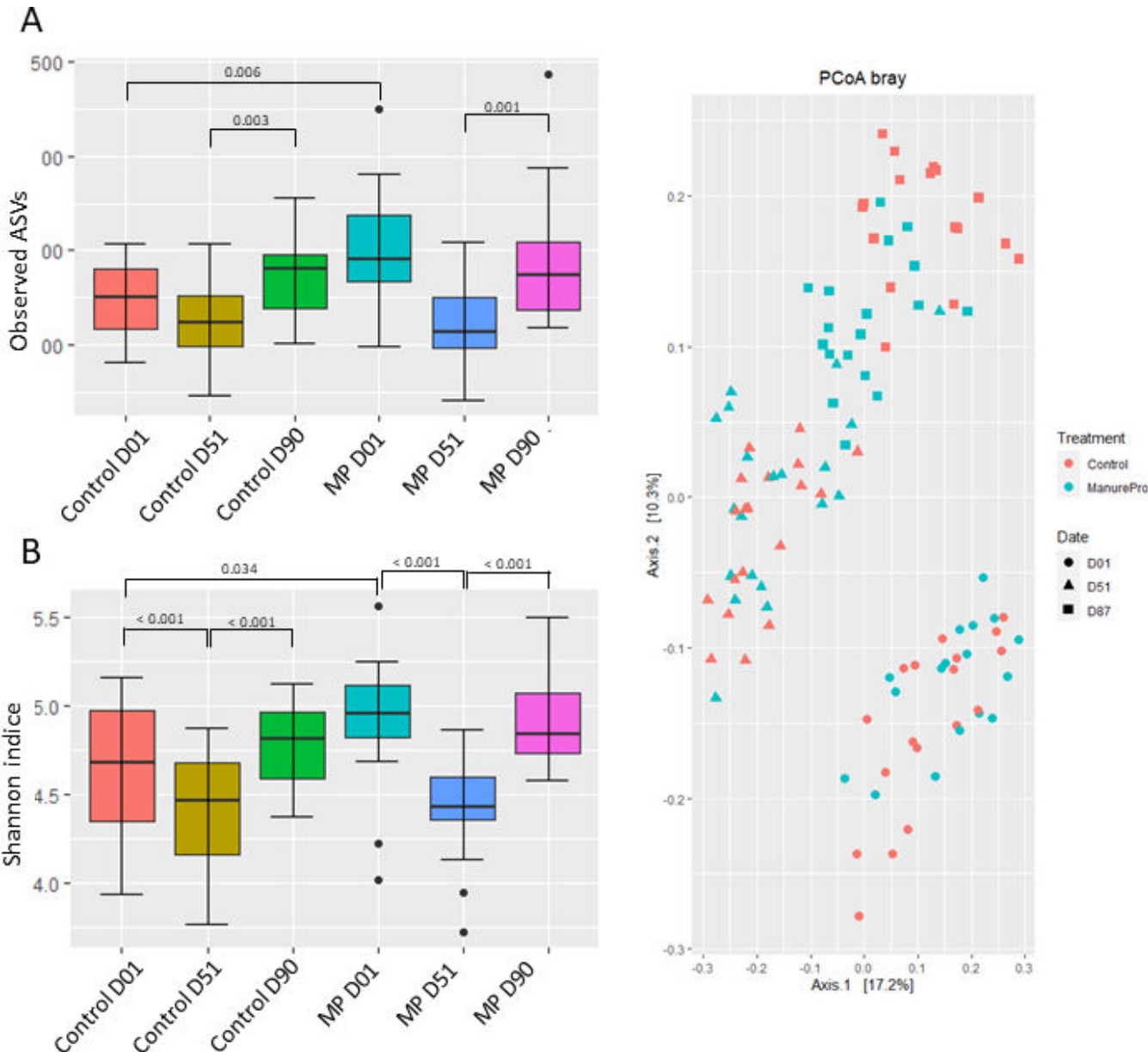

**FIG 5** Alpha diversity indices expressed through (A) observed ASVs and (B) Shannon indice and beta diversity expressed through (C) PCoA plot based on Bray–Curtis distance for normalized abundance data of Control and MP teat samples at D01, D51, and D90 (*n* = 18).

Control, indicating a less favorable environment for opportunistic microbial communities.

Bacterial communities were impacted by MP application with a higher diversity observed in MP beddings. It is commonly admitted that a high microbial diversity increases the robustness of a given environment after a disturbing event, as defined by its resistance, resilience, and functional redundancy (38). A high microbial diversity indicates a wide ecological niches occupation and limits the possibility for allochthonous microorganisms to establish and proliferate (39). The loss of diversity observed in case of dysbiosis generally allows a parallel growth of potential pathogenic microorganisms (40). Our results agree with this observation as differential analysis significantly associated several potential pathogens with Control groups, while MP-bedding groups were mainly associated with environmental and ruminant digestive bacteria (*Mesorhizobium*, Lachnospiraceae NK3A20, etc.). More precisely, bacterial genera such as *Streptococcus*,

**TABLE 4** Number of taxa and taxa of potential pathogens identified through Indicspecies analysis, *P*-values associated and relative abundances observed for teat samples from Control and MP groups analyzed at D01, D51, and D90

| Group | Number of identified taxa | Taxa of potential pathogens | *P*-value | Control (%) | MP (%) |
|---|---|---|---|---|---|
| Control D01 | 11 | *Lactococcus* (A = 0.47; B = 1) | 0.001 | 2.15 | 0.41 |
| | | *Latilactobacillus* (A = 0.80; B = 0.55) | 0.001 | 0.14 | 0.02 |
| | | *Secundilactobacillus* (A = 0.66; B = 0.44) | 0.002 | 0.13 | 0.02 |
| Control D51 | 26 | NA | NA | | |
| Control D90 | 47 | *Bifidobacterium* (A = 0.31; B = 1) | 0.006 | 1.18 | 0.43 |
| | | *Trueperella* (A = 0.33; B = 0.94) | 0.002 | 0.18 | 0.07 |
| MP D01 | 76 | NA | NA | | |
| MP D51 | 17 | *Acinetobacter* (A = 0.27; B = 1) | 0.001 | 11.33 | 10.99 |
| MP D90 | 54 | *Pseudomonas* (A = 0.31, B = 1) | 0.001 | 5.01 | 7.34 |

*Acinetobacter,* and *Staphylococcus* were identified in higher relative abundance and prevalence in the Control group over the trial. In a meta-analysis, Krishnamoorthy and colleagues have notably estimated the prevalence of *Staphylococcus* and *Streptococcus* species in subclinical or clinical bovine mastitis at, respectively, 28% and 12% worldwide over the 1979–2019 period (18). In a large-scale study, *Staphylococcus*, *Streptococcus,* and *Acinetobacter* were identified in 39.03%, 11.01%, and 3.38% of the 1,153 mastitis milk samples analyzed (41). *Corynebacterium*, a recognized genus involved in mastitis cases (42, 43), was observed in higher relative abundance in Control group D90, while no potential pathogen was identified in MP samples through IndicSpecies or LEfSe analyses. Only *Glutamicibacter* was observed to be enriched in MP D51 samples. This commensal genus is commonly observed in milk (44) and is known to produce short-chain fatty acids and may play a beneficial role in protecting the mucosal barrier and stimulating the host immune response (45).

Other potential pathogens, not always clearly associated with mastitis, have also been associated with Control group. *Fusobacterium* is an opportunistic pathogen, causing significant health issues in cattle such as liver abscesses and foot rot (46). *Fusobacterium necrophorum* has been identified in bovine summer mastitis (47), and in a Danish study, *F. necrophorum* has been isolated among other pathogens in 52% of the milk samples tested (48). *Legionella pneumophila* is a well-known cause of respiratory infections in humans. *Legionella* infections have been rarely documented in animals, and cattle is only considered accidental host for Legionellae (49). The low relative abundance of *Legionella*

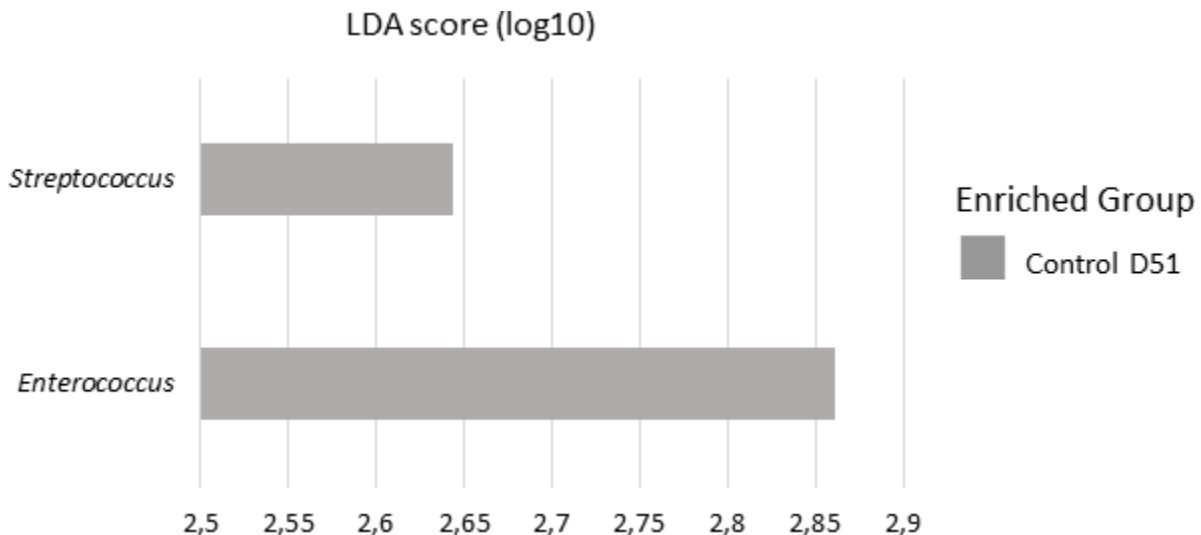

**FIG 6** LEfSe analysis identifying specific genera on Control (gray bars) and MP (blue bars) teat skin samples at D01, D51, and D90 (LDA cutoff 2.5, *P* < 0.05). Only genera associated with Control D51 were identified.

**TABLE 5** Total bacteria ($\log_{10}$ CFU/mL) and aerobic spores (counts/mL) measured in milk from Control or MP groups at D01, D51, and D90 ($n = 20$)

| | Control | | | MP | | | | P-values | | |
|---|---|---|---|---|---|---|---|---|---|---|
| | D01 | D51 | D90 | D01 | D51 | D90 | SEM | D01 | D51 | D90 |
| Tota bacteria ($\log_{10}$ CFU/mL) | 3.94 | 4.27 | 4.30 | 4.05 | 4.51 | 4.22 | 0.061 | 0.273 | 0.256 | 0.981 |
| Aerobic spore (counts/mL) | 14.06 | 10.00 | 11.18 | 14.18 | 15.59 | 12.65 | 1.973 | 0.850 | 0.850 | 0.506 |

DNA in bedding samples in our study is probably linked to environmental contamination through ground or potable water, for example (50).

The mechanisms by which MP product can modify bacterial profile of bedding remain not fully understood at this stage. *B. velezensis* has been used in agricultural industry, as an alternative to chemical fertilizers or pesticides, for its capacity to degrade toxic by-products or lignocellulose, and even for its bio-emulsifying properties (33). The functional inference analysis identified the enrichment of nitrifier denitrification pathway in MP group at D90. The nitrifier denitrification pathway leads to the transformation of ammonia ($NH_3$) into nitrite ($NO_2^-$) anf then nitric oxide (NO), nitrous oxide ($N_2O$), and finally molecular nitrogen $N_2$ (51). Dramatic environmental impacts of $NH_3$ are well documented (acidification and eutrophication of ecosystems, biodiversity shifts, etc.), and limitation of $NH_3$ emission is an important goal to achieve in agricultural industry (52). It has been demonstrated that microaerophilic conditions coupled with low organic matter contents and low pH stop the nitrifier denitrification pathway at the level of $N_2O$ production, a greenhouse gas with important consequences for environment (51). In our study, bedding samples presented both alkaline pH and high organic matter (OM) values, conditions more prone to promote the nitrifier denitrification pathway, which would, thus, limit $NH_3$ emission and lead to the ultimate transformation into $N_2$. Syringate degradation pathways was also observed to be enriched in MP D90. Syringate is a major product in the microbial and chemical degradation of syringyl lignin, with its degradation leading to the production of pyruvate and oxaloacetate (53). Several *Bacillus* strains have been observed among the lignolytic microorganisms identified in plants, soil, wood, or the gut (54); thus, the increase of syringate degradation pathway might be linked to the spraying of *Bacillus* through MP application. In addition, two L-arginine degradation pathways producing a biogenic amine (i.e., putrescine) as intermediate metabolite (55, 56) were enriched in Control group at D51. Putrescine can be a precursor for other biogenic amines production (spermine, spermidine), and its ingestion has been associated to adverse effects on animal such as decreased DMI and milk production of dairy cows (57), decreased intake, volatile fatty acids production, and total tract DM digestibility in steers (58) or lower growth rate in goat kids (59).

## Use of MP conditioner on RMS-bedding has beneficial impacts on teat skin microbiota

As teats of dairy cows are in direct contact with bedding several hours per day, a focus was made on the impact of MP use on bedding on the bacterial population of teat skin. Teat skin bacterial populations were affected by time, a result previously observed in other studies (60, 61). Noteworthy, Hohmann and colleagues observed that an increased frequency of bedding cleaning was associated with lower pathogens load on teat skin (61). Similar to the observations on bedding, a higher bacterial diversity was observed in teat skins of animals from MP group over the trial and it can be hypothesized that the more diverse population of commensal bacteria on teat skin would limit pathogen colonization and proliferation. This observation is strengthened by differential analysis which identified a higher number of several potential mastitis pathogens in Control than in MP group. At the start of the trial, teat skin of Control animals was characterized by presence of LAB (*Lactococcus*, *Latilactobacillus*, *Secundilactobacillus*). Among them, *Lactococcus* genus have been previously identified in bovine mastitis (62). At D90, almost all samples from Control group harbored one taxa assigned to *Truperella*. *Truperella* is opportunistic pathogen causing important diseases in domestic animals, and *T. pyogenes*

was isolated in approximately 25% of the mastitic cows quarters over a 2-year period (63). The LEfSe analysis identified the potential mastitis pathogens *Enterococcus* and *Streptococcus* in Control D51 teat skin samples. Among *Enterococcus* genus, *Enterococcus faecalis* and *E. faecium* have been isolated from 18% and 1.75% of 2,000 mastitis milks, respectively (64) and *E. faecalis* abundance have been shown to dramatically increase in mastitis milks compared to normal raw milk (65). Only two potential mastitis pathogens were associated with teat skin of MP group: *Acinetobacter* and *Pseudomonas*. Noteworthy, the prevalence of *Acinetobacter* was higher in MP, but the relative abundance was lower compared to Control group. *Pseudomonas* has been observed in some bovine mastitis (66). However, these potential pathogens have been previously observed on teat skin from healthy cows as well (60, 67) and specific qPCR or shotgun sequencing analyses would have been required to detect virulence factors linked to mastitis and, thus, confirm the lower pathogenic load in one of the experimental group.

## MP bacteria are not disseminated into environment and have no effect on milk yield or composition

Surprisingly, the sprayed bacteria from MP product (*B. velezensis*, *P. acidilactici,* and *P. pentosaceus*) were not significantly associated with MP beddings and *Bacillus* taxa were even associated with the Control group. However, due to high phylogenetic proximity among *Bacillus* species, taxa were assigned only at genus level, leading to an impossible discrimination and tracking of the commercial *Bacillus* strain used in this study. Similarly, *Pediococcus* taxa was identified at genus level and was observed only in MP group D51 at very low relative abundance (0.019%). As *Pediococcus* was not detected at D01, it seems that the genus was not part of the initial RMS bacterial population or was undetectable due to an extremely low abundance. When the application rate of MP decreased (from D31 to D90), *Pediococcus* was not observed anymore in MP group, suggesting a clearing effect and no perennial establishment of this species in the bedding environment and confirming thus the interest of a regular product application.

The absence of *Bacillus* transfer to teat skin and to milk was carefully assessed as this spore former genus is of great concern in dairy industry, being responsible for product defects (34). As previously observed in bedding samples, no *Bacillus* ASV was significantly associated with teat skin samples from MP group, while the relative abundance was higher in Control than MP group (on average 0.29% and 0.18%, respectively). *Pediococcus* was detected in very low relative abundance only in teat skin samples from MP D51.

No difference in milk yield or milk quality was observed between the two groups of animals. Milk total flora and aerobic spore counts were on average $4.25 \pm 3.53$ $\log_{10}$ CFU/mL and $12.94 \pm 0.85$ spores/mL, respectively, over the trial. These values are in lines with literature (68–70) and confirm the absence of negative impact of MP utilization on RMS-bedding on milk. The commercial bacteria applied to RMS-bedding were, thus, transferred neither to teat skin nor to milk over the 3 months trial.

## Use of MP conditioner on RMS-bedding increased cow's health parameters

Indirect beneficial effects of MP conditioner use on RMS-bedding were observed on several animal health parameters. Soiled udder being one of the main factors triggering mastitis development, a specific attention was given to animal cleanliness. After 51 days of MP utilization on bedding, the animals were characterized by a better hygiene (cleanliness score) and a lower load of bacteria on teat skin, probably linked to the higher DM observed on MP-bedding. This was also reflecting on SCC in milk. In clinical or sub-clinical mastitis, pathogens invade mammary ducts and proliferate. The inflammatory response results notably in white blood cells infiltration to eliminate the pathogenic bacteria (71). These immune cells are identified as SCC in milk and a threshold of 200,000 cells/mL (i.e., 5.3 $\log_{10}$ cell/mL) of milk is commonly accepted to characterize an infected cow (72). In our study, from D44 to D76, SCC in milks from Control group were above this threshold and on average 33.67% higher (5.61–5.49 $\log_{10}$ SCC/mL) than in MP group (5.13–5 $\log_{10}$ SCC/mL), indicating that cows from MP group presented lower activation

of their immune system reflecting the lower exposition to mastitis pathogens. Several SCC thresholds are applied by different countries to determine the lowest milk quality acceptable, ranging from 400,000 cells/mL in EU to 750,000 cells/mL in the USA for bulk tank milk (73–75). High SCC milks have been shown to adversely affect cheese production or pasteurized milk shelf life (76). More specifically in France, a threshold of 250,000 cells/mL has been set above which financial penalties are applied for the farmer (77). In this study, significantly more animals presented milk samples above this limit in Control than in MP group, highlighting the potential economical interest of controlling bedding quality and its further potential benefits on animal health. Mastitis pathogens can be classified into contagious or environmental categories with contagious bacterial species such as *Streptococcus agalactiae* and *Staphylococcus aureus* being spread from an infected cow to a healthy one, usually at milking time (78). Environmental bacteria (*E. coli*, *Klebsiella*, environmental Streptococci, etc.) come from the cow environment, and contrary to contagious mastitis pathogens, it is not possible to totally eliminate them due to their endemic nature. Management practices aiming to improve cleanliness of the animals and their surroundings are the best strategies to control environmental mastitis outbreaks (5, 78). Over the trial, the number of effective environmental mastitis pathogens identification was higher in Control group indicating that the use of MP on RMS-bedding led to a better animal hygiene and health and subsequently decrease the potential risk of mastitis onset.

## Conclusion

The use of MP has shown to improve both physico-chemical and bacterial quality of RMS-beddings over a 3-month trial. MP bedding samples were significantly less associated with potential mastitis bacteria. This greater sanitary quality had further impact on animal health as teat skin of animals in MP group was less soiled and presented an improved bacterial composition with less potential mastitis pathogens than in Control animals. Finally, milk quality was not affected by MP application onto RMS-bedding, confirming the absence of bacterial transfer from the commercial product to the milk. Further research is still needed to understand the mechanisms behind the modifications of RMS-bedding bacterial composition observed in this study, and the respective roles of each component of the MP product.

## MATERIALS AND METHODS

### Animal housing

The study was conducted on a commercial farm in the Czech Republic. The building was divided into 8 pens with 4 pens (2 on the north and 2 on the south side) dedicated to the trial with an equal repartition between pens submitted to Control or MP treatment. Each pen was composed of 68 cubicles of 2.5 m² and one milking robot (LELY Holding N.V., Maassluis, Netherlands). Bedding in the cubicles was composed of Recycled Manure Solids (RMS) previously prepared from fresh manure from the same farm. Once a week, about 22 kg of fresh RMS was added per cubicle and compacted mechanically by tractor passage. Twice a day, the back part of the cubicles was manually scraped to remove organic waste. This experiment was conducted between the 31st of May and the 1st of September 2021. Temperature of the building farm was ranging between 12.7°C ± 2.9°C and 20.2°C ± 4°C.

### Animal selection

A total of 228 Holstein and Simmental crossbreed Holstein cows were enrolled in the Control ($n = 115$) and MP (Manure Pro, $n = 113$) groups, with parity ranging from 1 to 8 and with averaged day in milk (DIM) of 183 ± 119 at the start of the trial. No

significant effect of parity or DIM was observed between the two experimental groups (*t*-test, *P*-value > 0.05).

## Bedding experimental treatments

In the treated group (MP), the bedding conditioner MANURE PRO (Lallemand SAS, Blagnac, France), composed of live *Bacillus velezensis*, *Pediococcus acidilactici,* and *P. pentosaceus* at $5 \times 10^9$ CFU/g and cellulolytic enzymes, was sprayed with a backpack sprayer once a week at a concentration of 1 g/m$^2$/week for 1 month in the pen, cubicles, exercise area, and corridor. During the following 2 months, the application rate was decreased to 0.5 g/m$^2$/week as the 1-month set-up phase was complete. The application rate from D01 to D30 was set up at twice the commercial dosage used afterward in order to prime the effect of the sprayed inoculum on bedding environment. Building surfaces of Control group did not receive any treatment.

## Sampling procedures

RMS, bedding, and teat skin sampling were done before the first application, at the middle and end of the trial, respectively, Day 01, Day 51, and Day 90. The bedding samples were carefully taken with clean gloves at 5 cm of depth in four locations in the cubicle (two front, two back), avoiding feces. Bedding samples were mixed thoroughly and stored at −80°C until processing. Two replicates were collected per row of cubicles, in three rows per pen, for a total of six bedding samples per pen per time. Teat skin was sampled using sterile swabs, pre-soaked with AMIES medium (FLMedical, Torreglia, Italy), by gently rubbing two opposite teats of the animal. Swabs were kept in buffer according to manufacturer recommendations and stored at −80°C. Teat skin samples were collected on 10 animals per pen (20 per experimental group) balanced in terms of rank of lactation (parity 2 and 3) and with DIM ranging from 70 to 171.

## Sample analysis

### Bedding

The pH of bedding samples was measured with a pH meter and 10 g of fresh bedding diluted into 50 mL of $CO_2$-free distilled water. DM and OM were analyzed according to the Unified Work Procedures issued by the Central Institute for Supervising and Testing in Agriculture (Czech Republic) based on Commission Regulation (EC) no, 152/2009 (79). Briefly, 50 g of fresh bedding was dried for 4 h at 103°C before weighing for DM calculation. OM was analyzed by ashing 1 g of dried sample at 550°C. The measurements were done by the laboratory S.O.S. Skalice n. Svit., s.r.o., Czech Republic.

### Milk

Milk production was followed every day for each cow individually. Milk conductivity (mS/cm) was measured automatically by milking robots at each milking. Somatic cells count (SCC) was individually followed every 15 days (six measures over the experimental period) by flow cytometry using a Combi Foss (Foss Electric, Hillerød, Denmark).

### Microbiology

Swab samples from teats were thoroughly mixed into their respective buffer, and an aliquot was used for total flora numeration. Milk total culturable bacteria was determined by plating an aliquot of raw milk from 20 cows per group. Aerobic spores were enumerated by heating 1 mL of milk at 80°C for 10 min before enumeration. All microbiology analyzes were done according to ISO 4833-1:2013 (80) at the State Veterinary Institute Jihlava, Czech Republic.

## Mastitis detection

Based on conductivity measurements provided by the Lely robot at each milking, any cow milk above 90 mS/cm was classified as "mastitis suspicion," while cows below the threshold were classified as "healthy." Milk samples from the "mastitis suspicion" category were sampled and tested through a MicroMastTM system (MicroMast, Světlá nad Sázavou, Czech Republic) according to manufacturer recommendations. This on-farm method of mastitis diagnostics allowed the identification of pathogens in milk samples among the main Gram-negative and -positive bacteria responsible for mastitis.

## Cleanliness score

Cleanliness score was assessed by a trained staff member following the grid provided by the NZ dairy organization (www.dairynz.co.nz). A score ranging from 0 (clean) to 2 (soiled) was assigned to the back, flank, tail, hind leg, and udder of each animal with 6 points being the maximum and worst hygienic condition.

## Molecular analysis

### DNA extraction

DNA extraction was performed on 4 g of frozen bedding samples (−80°C) or from teat skin swabs with the Quick-DNA Fecal/Soil Microbe 96 kit (Zymo Research, Irvine, CA, USA) according to manufacturer's instructions. Bedding samples and swabs were blended with 20 mL of PBS in stomacher bags with 280 µM filter, for 1 and 5 min, respectively, using a stomacher (Smacher, bioMérieux, normal speed). For bedding samples, 3 mL of suspensions were first centrifuged at 28 g for 30 s, supernatant was collected and centrifuged at 10,000 $g$ for 1 min. For swab samples, suspensions were centrifuged at 4,122 $g$ for 30 min. Pellets were collected and stored at −20°C until DNA extraction. DNA yield and quality was determined with a Nanodrop 1000. DNA extracts were stored at −20°C until analysis.

### Amplicon sequencing analysis

Bacterial diversity and taxonomic composition were analyzed by 16S rRNA gene amplicon sequencing. The hypervariable V3–V4 regions of the 16S rRNA gene were targeted using the primers set 341F 5′-CCTACGGGAGGCAGCAG-3′ and 806R 5′-GGAC-TACNVGGGTWTCTAAT-3′ (81, 82). A total of 194 samples were sequenced on a MiSeq high-throughput sequencer (Illumina, San Diego, CA) at the GeT-PlaGe facility (INRAE, Genotoul, Toulouse, France). MiSeq Reagent Kit v3 was used according to the manufacturer's instruction (Illumina Inc., San Diego, CA) and paired-end 250 bp sequences were generated. The 16S rRNA gene sequences were processed the DADA2 pipeline 1.22.0 (83) for the pipeline's steps of filtering, trimming (Forward reads trimmed at 235 bp, Reverse reads trimmed at 220 bp), dereplication, sample composition inference, and chimera removal. The Silva nr v.138.1 database (84) was used to assign taxonomy to the ASVs with a minimal boostrap of 30. Diversity analyses were performed with the Phyloseq 1.42.0 R package (85). The sequencing run produced a total of 4,462,388 reads that were used as input for bioinformatic analysis, and a total of 3,218,612 reads were kept after trimming, chimeral removal, and filtration steps. Rarefaction curves were analyzed to confirm the correct depth sequencing of each sample.

### Sequencing data analysis

Raw ASV abundances were used for the composition analysis. Alpha diversity analysis was performed using data rarefed down to the lowest library sampling size with the Phyloseq 1.42.0 R package (85), while beta diversity analyses used transformed abundance table with the DESeq2's variance stabilizing transformation (86). The overall dissimilarity of the microbial community across days and groups was evaluated by

principal coordinates analysis (PCoA) based on Bray-Curtis dissimilarity. The significance of differences between experimental groups was tested by analysis of similarity (ANOSIM) and multiple comparisons using Benjamini and Hochberg correction. Indicator species in bedding and on teat skin were identified using the "Indicspecies" package in R (87, 88). This method identifies species that are specific to one group with high fidelity (most samples in that group have the species). For this, a genus-level table was used as input. Each genus ecological niche preference (Control or MP at each sampling day) was identified using the Pearson's phi coefficient of association (corrected for unequal sample sizes) using 999 permutations. All samples were considered independent. LEFSe (Linear Discriminant analysis Effect Size) analysis was performed on bedding and teat skin samples using RLE (Relative Log Expression) normalization with a 2.5 LDA score cutoff on taxa tables agglomerated down to the genus level and a minimum of detection fot his genus set at 5 samples per group. Functional inference analysis was performed on 16S bedding data following the PICRUSt2 v2.5.0 pipeline (89) with HMMER (http://hmmer.org), EPA-NG (90), gappa (91), and SEPP (92) tools for phylogenetic placement of reads; castor tool for hidden state prediction (93); and MinPath tool for pathway inference (94). Statistical analysis was then done using STAMP 2.3.1 software with two groups comparison through two-sided Welch $t$-test and significance was declared at $P < 0.05$.

## Statistical analysis

Data were analyzed using GraphPad 10.0.0. Microbiological data and SCC were $\log_{10}$-transformed before statistical analysis. Changes in bedding parameters were analyzed through the non-parametric Mann-Whitney test considering the evolution between D01 and D51 for Period 1 and between D51 and D90 for Period 2. Mixed model considering Group and Time as fixed factors and the effect of their interaction was applied on animal hygiene parameters, SCC and milk production. Cow was set as random factor and was first nested within the pen to account for any potential pen effect. The absence of a pen effect on any of the tested parameters led to keep only the cow as random factor. Milk production was averaged for each animal per month of treatment application to follow the difference in MP application rate with Month 1 from D01 to D30 at 1 g/m²; Month 2 from D31 to D60; and Month 3 from D61 to D90, both at 0.5 g/m². Fischer test was applied on suspicious mastitis cases observed in both groups per week and over the trial, as well as on the number of animal presenting SCC in milk above 250,000 cells/mL. Number of suspicious mastitis leading to effective pathogen identification through on-farm plating in each group were compared through non parametric Mann-Whitney test. Mann-Whitney test was also applied to total milk bacteria and milk aerobic spores data. For all statistical analysis, significant difference was declared at $P < 0.05$.

## ACKNOWLEDGMENTS

The authors would like to thank the staff from Probionic s.r.o. farm for their help in the animal experiment, and Dr. Chaucheyras-Durand for her valuable help in reviewing this manuscript.

J.P. and B.F. designed and supervised the project with contributions from E.C. C.A. performed the DNA extraction. L.D. performed sequence data analyses. B.F. and L.D. performed the statistical analyses. L.D. wrote the manuscript and prepared the figures. All the authors read and approved the final manuscript.

B.F., L.D., E.C., C.A., and J.P. are employees of Lallemand SAS company. Other authors declare no competing.

## AUTHOR AFFILIATION

[1]Lallemand SAS, 19 rue des Briquetiers, Blagnac, France

## AUTHOR ORCIDs

Lysiane Duniere  http://orcid.org/0000-0003-2918-896X

## FUNDING

| Funder | Grant(s) | Author(s) |
|---|---|---|
| Lallemand (Lallemand Inc.) | | Lysiane Duniere |
| | | Bastien Frayssinet |
| | | Caroline Achard |
| | | Eric Chevaux |
| | | Julia Plateau |

## AUTHOR CONTRIBUTIONS

Lysiane Duniere, Data curation, Formal analysis, Methodology, Validation, Visualization, Writing – original draft | Bastien Frayssinet, Conceptualization, Data curation, Investigation, Software, Writing – review and editing | Caroline Achard, Methodology, Writing – review and editing | Eric Chevaux, Conceptualization, Supervision, Validation, Writing – review and editing | Julia Plateau, Conceptualization, Funding acquisition, Writing – review and editing

## DATA AVAILABILITY

The sequence datasets generated during the current study are available in the SRA repository, (https://www.ncbi.nlm.nih.gov/sra/) as PRJNA1031159. All the other data are included in this published article and its supplementary information files.

## ADDITIONAL FILES

The following material is available online.

### Supplemental Material

**Supplemental material (Spectrum04263-23-S0001.xlsx).** Result tables of IndicSpecies. analysis on RMS-bedding samples in both Control and MP groups
**Supplemental Figures S1 to S3 and supplemental Table S1 (Spectrum04263-23-S0002.docx).** Supplemental figures and tables.

### Open Peer Review

**PEER REVIEW HISTORY (review-history.pdf).** An accounting of the reviewer comments and feedback.

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
