## [Reviewer comments · Microbiology Spectrum]

Microbiology Spectrum

Conditioner application improves bedding quality and bacterial composition with potential beneficial impacts for dairy cow's health.

Lysiane Duniere, Bastien Frayssinet, Caroline Achard, Eric Chevaux, and Julia Plateau

Corresponding Author(s): Lysiane Duniere, Lallemand SAS

Review Timeline:

Submission Date:

January 16, 2024

Accepted:

January 29, 2024

Editor: Jeffrey Gralnick

Reviewer(s): The reviewers have opted to remain anonymous.

Transaction Report:

DOI: <https://doi.org/10.1128/spectrum.04263-23>

Re: Spectrum04263-23 (Conditioner application improves bedding quality and bacterial composition with potential beneficial impacts for dairy cow's health.)

Dear Dr. Lysiane Duniere:

Based on your response and revisions from the prior round of review, your manuscript has been accepted, and I am forwarding it to the ASM production staff for publication. Your paper will first be checked to make sure all elements meet the technical requirements. ASM staff will contact you if anything needs to be revised before copyediting and production can begin. Otherwise, you will be notified when your proofs are ready to be viewed.

Sincerely,
Jeffrey Gralnick
Senior Editor
Microbiology Spectrum